# Osteoporosis Evaluation by Radiofrequency Echographic Multispectrometry (REMS) in Primary Healthcare

**DOI:** 10.3390/diagnostics15070808

**Published:** 2025-03-22

**Authors:** Ana Vieira, Rute Santos

**Affiliations:** 1Local Health Unit of Guarda, 6300-858 Guarda, Portugal; aanaiisabelsv@gmail.com; 2Medical Imaging and Radiotherapy Department, Coimbra Health School, Polytechnic University of Coimbra, 3045-093 Coimbra, Portugal; 3H&TRC-Health & Technology Research Centre, Coimbra Health School, Polytechnic University of Coimbra, 3045-093 Coimbra, Portugal; 4Interdisciplinary Centre for the Study of Human Performance, University of Coimbra, 3004-531 Coimbra, Portugal

**Keywords:** osteoporosis, primary healthcare, radiographer, radiofrequency echographic multispectrometry (REMS), ultrasound, bone mineral density (BMD)

## Abstract

**Background/Objectives**: Radiofrequency echographic multispectrometry (REMS) technology has emerged as a promising alternative for osteoporosis diagnosis. This non-ionising, portable and accessible method enables early detection of osteoporosis in primary healthcare settings. The aim of this study was to assess the effectiveness of REMS in evaluating osteoporosis within primary healthcare. **Methods**: Bone mineral density was assessed in 86 participants trough 172 scans of the lumbar spine and femur, using REMS technology in two Portuguese primary healthcare units in Guarda. **Results**: In the lumbar spine evaluation, 51.2% of the participants had osteopenia and 31.4% osteoporosis; in the femur evaluation, 43.0% had osteopenia and 34.9% osteoporosis. The data indicated a significant prevalence of bone fragility. The bone mineral density estimated by radiofrequency echographic multispectrometry showed good agreement with the clinical diagnosis, suggesting that this technology is effective in the early detection of osteoporosis. **Conclusions**: Bone densitometry using REMS method, performed by a radiographer in primary healthcare settings, offers a viable and innovative alternative for the effective detection of osteoporosis and osteopenia.

## 1. Introduction

According to the World Health Organisation (WHO) and scientific evidence, health systems based on primary healthcare have better health outcomes and represent the first contact between the user and the health service [1,2]. Primary healthcare provides comprehensive care close to the community and is fundamental to promoting health and preventing disease [2]. Health professionals working in this area play an essential role in raising awareness among the population of the importance of early detection, encouraging people to seek medical care at the first sign and/or symptom of concern and promoting participation in screening programmes, thus ensuring that the population receives the necessary treatment as early as possible [3,4,5]. Primary healthcare is also essential for preventing osteoporosis and a better therapeutic outcome, as it identifies risk factors and the disease at an early stage [6,7,8].

Osteoporosis is a common and silent disease, characterised by a decrease in bone mass and alterations in the microarchitecture of bone tissue, which results in increased bone fragility and greater susceptibility to fractures [9,10]. This condition has a significant impact on the health and well-being of patients, increasing morbidity and mortality. Known as the “silent thief”, osteoporosis progresses without symptoms until fractures occur, making prevention and early diagnosis essential to reduce complications and the costs associated with them [11,12,13].

Osteoporosis and the fractures that result from it are currently one of the biggest public health problems. More than 500,000 Portuguese are estimated to have osteoporosis, a number that is increasing due to the ageing population caused by increasing life expectancy and declining mortality rates [14,15]. Adopting strategies that promote an increase in average life expectancy healthily is imperative. In recent years, there has been growing interest in identifying new parameters and technologies that make it possible to diagnose osteoporosis and predict the risk of fractures in a simple and economically sustainable way, given that the cost of fractures in Portugal accounts for 5.6% of health expenditure, one of the highest percentages in Europe [15,16,17].

The gold-standard imaging method for diagnosing osteoporosis, standardised in Portugal, is bone densitometry by dual-energy X-ray absorptiometry (DXA), which is easy to perform and allows for the quantitative analysis of bone mineral density [18,19,20]. However, this method has disadvantages, such as the use of ionising radiation, the need for users to travel to imaging services and limitations in cases of structural anomalies, such as osteoarthritis, which can distort the results as it will artificially increase bone mineral density measurements in the lumbar spine [19,21,22].

Early detection of individuals with low bone mineral density using bone densitometry and/or risk factor assessment is crucial for initiating appropriate treatment, reducing the incidence of osteoporotic fractures and managing bone health, particularly in at-risk populations such as postmenopausal women and older adults [23,24]

Some risk factors are difficult to quantify and vary in importance, making assessing each patient’s overall risk of osteoporosis complicated. In addition, DXA’s limitations, such as its cost and availability, have led to exploring alternative screening strategies [23,25].

Recently, a new approach to diagnosing osteoporosis has emerged: Radiofrequency Echographic Multispectrometry (REMS), represented by the EchoStation equipment and approved by the European Society for Clinical and Economic Aspects of Osteoporosis, Osteoarthritis and Musculoskeletal Diseases [26].

This innovative technology makes it possible to assess bone mineral density in the spine and femur and predict the risk of fractures independently of bone mineral density without using ionising radiation. It can be applied to specific groups, such as children and pregnant women. It is also easily transportable, promoting faster and more convenient diagnosis in primary healthcare and economic sustainability [27,28,29,30,31].

REMS technology uses an echographic device with a convex transducer operated at 3.5 MHz. This method fully exploits all the spectral characteristics of the ‘raw’ radiofrequency (RF) (unfiltered signals) acquired during an echographic scan of the target anatomical site. This analysis of the unfiltered signals allows for the retention of as much information about the bone tissue as possible, enabling the acquisition of information in terms of both quantity and quality [26,28]. The data obtained are synthesised into a spectrum compared with reference spectral models (including ‘normal’ and ‘osteoporotic’ individuals), considering variables such as gender, age, race and body mass index [26].

The automatic combined analysis of B-mode images and corresponding RF data provides the following parameters: Osteoporosis Score (OS) which is directly correlated with bone mineral density (BMD), T-Score and Z-Score, which are calculated by comparisons with National Health and Nutrition Examination Survey reference curves, and the Frailty Score (FS), which quantifies bone strength by assessing structural frailty independently of BMD [27,28].

Several multicentre studies have clinically validated the application of REMS technology for the diagnosis of osteoporosis, demonstrating a high correlation with DXA in the assessment of bone mineral density and high accuracy in the diagnosis. Studies such as those by Di Paola et al. (2019) and Nowakowska-Płaza et al. (2021) confirmed the high diagnostic agreement between REMS and DXA [29,30]. In addition, investigations such as those by Cortet et al. (2021) and Amorim et al. (2021) have shown that REMS has a sensitivity and specificity of over 90%, highlighting its excellent ability to discriminate between normal and osteoporotic individuals [27,31]. Studies focused on pathologies associated with secondary osteoporosis and patients with degenerative diseases and bone diseases have further strengthened the potential of REMS, demonstrating its reliability even in the presence of artefacts, particularly useful in diagnosing osteoporosis in the elderly [28,32].

Despite these advances, implementing screening programmes for osteoporosis still presents challenges. One significant barrier is the underutilisation of screening tools, especially among high-risk populations. Studies show that many people who fulfil the screening criteria, such as postmenopausal women and the elderly, do not undergo DXA scans or other forms of screening [33,34,35].

According to the SCOPE 2021 report, the DXA scan costs 35 euros in Portugal, and the user is fully reimbursed [14,15], however, the qualitative study in Italy by Borsoi et al. associates the REMS method with lower costs than DXA. These results help policymakers understand the value of REMS technology in the early diagnosis of osteoporosis and support their decision regarding reimbursement and dissemination of the technology [36].

Therefore, due to the lack of DXA equipment in Portuguese primary care, it is believed that REMS technology can fill this gap, enabling bone health screening in these services [9,15]. This study aimed to contribute to the early detection of osteoporosis in primary healthcare. It uses portable equipment that is easy to transport and handle and does not use ionising radiation, making screening more accessible and practical.

## 2. Material and Methods

This cross-sectional analytical study used a non-probabilistic convenience sample of 86 participants voluntarily recruited from primary healthcare units in Guarda, Portugal. Data collection took place in June 2024. Inclusion criteria comprised individuals over 30, while exclusion criteria included those with prostheses in the assessed region and those with low adherence to the study’s participation requirements. The study was approved by the Board of Directors and the Ethics Committee of the Unidade Local de Saúde da Guarda (No. UI-NS-240033). Every participant signed an informed consent form for analysis and the General Data Protection Regulation processed personal data.

### 2.1. Exams and Equipment

All the participants completed a questionnaire based on a review of the scientific literature. The questionnaire included both modifiable risk factors (such as dietary habits, physical activity, alcohol and tobacco consumption, sun exposure and supplementation) and non-modifiable risk factors (such as age, gender, history of fractures, menopause and the presence of associated chronic pathologies).

Bone densitometry tests were then carried out on the spine and femur using REMS technology with the EchoStation device and EchoStudio software (Version No. 002-0000) (Echolight Spa, Lecce, Italy), registered with Informed under the medical device code 24150843. The same radiographer performed all the examinations.

The radiographer entered the participants’ data (age, sex, weight, height and age at menopause) into the device before the examination began. The machine has a 3.5 MHz broadband convex ultrasound transducer.

All participants were scanned in two anatomical regions: the proximal femur and the lumbar spine. For each scan, the radiographer adjusted the depth (ranging from 60 to 210 mm for the spine and from 90 to 150 mm for the femur) and the focus (ranging from 21 to 100 mm for the spine and from 21 to 60 mm for the femur).

#### Positioning the Probe

Proximal femur: The probe was positioned obliquely, parallel to the femoral head-neck axis. In the same ultrasound image, we visualise the bone line of the femoral head, femoral neck and trochanter. The bone interface of the femoral neck was positioned parallel to and below the reference line on the screen (red line). The acquisition took around 40 s, always with the translator in the same position, and processing took 1–2 min (Figure 1).

Lumbar spine: The probe was placed in the xiphoid appendix to visualise the first lumbar vertebra (L1), and then we followed with the probe in the umbilical plane to visualise and visualise L2 to L4. Different intensities of probe pressure were required to position the vertebra on the ultrasound image. The vertebral bone interfaces were located 1 to 2 mm below and parallel to the horizontal reference line (red line). The acquisition took around 80 s, with a further 1–2 min of processing (Figure 2).

The software detects bone interfaces and identifies regions of interest (ROIs) for diagnostic assessment. The key feature is the exploitation of B-mode images to identify target bone interfaces and ROIs, combined with the analysis of unfiltered radiofrequency signals.

The tests were all carried out with the participant in the supine position.

The acquired ultrasound data are processed using an automated algorithm that performs various spectral and statistical analyses. These data are synthesised into a spectrum and compared with reference spectral models while considering variables such as gender, age, race and body mass index. The analysis focuses explicitly on trabecular bone, ensuring that artefacts do not impact the results.

The analysis of both the ultrasound images and the native radiofrequency signals allows for the calculation of bone mineral density (BMD) in grams per square centimetre (g/cm^2^), as well as T-Scores and Z-Scores calculated using the National Health and Nutrition Examination Survey reference database.

The results are categorised based on the World Health Organisation (WHO) definitions:-T-Score ≥ −1 (normal);-T-Score between −1 and −2.5 (osteopenia);-T-Score ≤ −2.5 (osteoporosis) [17].

The entire procedure took 7 to 10 min, and every participant, as illustrated in Figure 3, received a printed final report of the tests.

Based on this report on bone densitometry using the REMS method of the lumbar spine (a) and the right femur (b) (Figure 3):

In Figure 3, the graph on the left of the spine and femur assessments is analysed. The black circle indicates the participant’s bone mineral density compared to the standard reference according to age and gender. The green band represents normal bone density, the yellow band indicates osteopenia (decreased bone density), and the red band indicates osteoporosis. The participant’s line is in the yellow zone, confirming the diagnosis of osteopenia in both assessments.

In the assessment of the lumbar spine in Figure 3, the graph on the right identifies each vertebra and its colour based on the T-Score (Red: Osteoporosis; Yellow: Osteopenia; Green: Normal). Each participant’s vertebra is yellow, confirming the diagnosis of osteopenia. Below the image are the specific BMD and T-Score values for the L1 to L4 vertebrae.

When assessing the right proximal femur in Figure 3, the graph on the right represents the femur, with the femoral neck coloured according to the T-Score value (Red: Osteoporosis; Yellow: Osteopenia; Green: Normal). The participant’s femoral neck is yellow, confirming the diagnosis of osteopenia. Below the image are the specific BMD values, T-Score, Z-Score for the trochanter and the value for the total hip.

It should be noted that BMD values are provided for each lumbar vertebra, femoral neck, total hip and greater trochanter. The REMS BMD measurement is based on spectral models initially derived from a reference population that underwent a DXA scan to define osteoporosis, which experienced operators double-checked to avoid possible errors (including incorrect positioning of the user, inaccurate data analysis, presence of artefacts) due to providing unreliable BMD values [26,27,28].

### 2.2. Statistical Analysis

After data collection, the data were coded and entered into a database using SPSS software, version 26.0. The descriptive analysis included frequency distributions, means, standard deviations, and graphical representations. For the inferential analysis, we utilised the Chi-squared test of independence to identify significant associations between participants’ characteristics and bone mineral density (BMD) results in the spine and femur. When the conditions for the Chi-squared test were not satisfied, Fisher’s exact test was used as an alternative. The ANOVA test was also employed to compare ages across the diagnostic groups: normal, osteopenia, and osteoporosis since the Shapiro–Wilk test confirmed that the distributions were normal. A significance level of 5% (*p* < 0.05) was established for determining statistical significance.

## 3. Results

The final analysis included 172 bone mineral density tests, 86 of which were for the lumbar spine and 86 of which were for the proximal femur.

Table 1 shows the patients’ sociodemographic, clinical and lifestyle characteristics included in the statistical analysis.

In Table 1, the average age of the participants was 62.60 ± 12.33 years, and the majority were female (87.2%); of the female participants, 76.0% were in the menopause phase, with the average age of onset of menopause being 48.89 ± 6.71 years. All the respondents were Caucasian. Of the study participants, 4.7% had osteosynthesis material in the pelvis or spine, 23.3% had a diagnosis of osteoporosis confirmed by their doctor, and 22.1% said they had bone fractures.

Table 2 shows the characterisation of health and eating habits:

The results of the bone mineral density assessment in the spine are displayed in [Fig diagnostics-15-00808-ch001]. The [Fig diagnostics-15-00808-ch002] presents the results of the bone mineral density assessment for the femur.

Table 3 presents the age results for each type of BMD outcome in the spine and femur.

Table 3 shows that the average age increases with the severity of the diagnosis. After checking the normality of the distributions and applying the parametric ANOVA test, we found that at least one of the groups had a significantly different mean score. By applying Scheffe’s multiple comparisons test, we found that the average age of the normal diagnosis significantly differed from that of the other diagnoses.

Table 4 shows the results of the cross-referencing of the bone mineral density assessment in the spine using the REMS method, with the doctor’s confirmed diagnosis of osteoporosis and the possible bone fracture, as well as the respective results of the chi-squared test of independence.

Looking at Table 4, we see that the participants classified with a normal spinal diagnosis had no confirmation of osteoporosis from their doctor. By applying the chi-squared test of independence (*p* = 0.046), we can see that the result of the REMS method presented is significantly associated with the diagnosis confirmed by the doctor. Regarding the existence of a bone fracture, by applying the chi-squared test of independence (*p* = 0.278), we found that the diagnosis of the REMS method presented was not significantly associated with a bone fracture.

Table 5 shows the results of the cross-referencing of the bone mineral density assessment of the femur using the REMS method, with the doctor’s confirmed diagnosis of osteoporosis and the possible bone fracture, as well as the respective results of the chi-squared test of independence.

Table 5 shows that the participants classified with a normal diagnosis of the femur had no confirmation of osteoporosis from their doctor. By applying the chi-squared test of independence (*p* < 0.01), we can see that the diagnosis of the femur from the REMS method presented is significantly associated with the diagnosis confirmed by the doctor. About the existence of a bone fracture, by applying the chi-squared test of independence (*p* = 0.278), we found that the diagnosis of the femur using the REMS method presented was not significantly associated with a bone fracture.

## 4. Discussion

The study stands out for its introduction of REMS technology in a primary care setting, exploring its potential as an innovative and accessible alternative for the early detection of osteoporosis.

The study found that the average age of participants with osteopenia and osteoporosis was significantly higher than in individuals with normal BMD (Table 3). The relationship between age and worsening osteoporosis is in line with scientific literature, which shows accelerated bone loss from the age of 50, especially in post-menopausal women, due to the drop in oestrogen levels [37,38]. The results of the BMD assessment in the spine (*p* = 0.034) and femur (*p* = 0.000) using REMS technology showed a significant association with menopause. These data corroborate the cross-sectional study by Oliveira et al. (2019), which showed a higher prevalence of osteoporosis in older people and women [39].

Body mass index (BMI) analysis revealed that half of the participants were overweight (50%) and 10.7% were obese (Table 1). The relationship between BMI and BMD is complex. However, obesity may be related to greater bone mineral density; it paradoxically increases the risk of fractures due to muscle weakness and the risk of falls. Furthermore, a BMI < 19 kg/m^2^ is an indicator of osteoporosis, thus emphasising the importance of BMI being a screening criterion [40,41,42].

Sun exposure and consuming foods high in vitamin D and calcium, essential for bone health, showed significant gaps. 68.6% of participants avoid sun exposure between 10 am and 4 pm, which can compromise vitamin D synthesis. In addition, 79.1% report eating vitamin D-rich foods only once or twice a week, and only 17.4% use vitamin D supplements. Calcium intake is also worrying, with 44.2% consuming these foods once or twice a week and only 19.8% more than five times a week (Table 2). Vitamin D deficiency is widely recognised as a risk factor for osteoporosis and is closely linked to calcium absorption and bone mineralisation. Insufficient sun exposure can aggravate this deficiency and puts a significant part of the population studied at risk of developing or aggravating osteoporosis [43,44]. The literature review by Voulgaridou et al. (2023) indicated that combined vitamin D and calcium supplementation improves BMD and decreases PTH activity, preventing the risk of fractures [45]. In addition, the literature emphasises the importance of other nutrients, such as magnesium, phosphorus and vitamin K, for bone health [43]. Although phosphorus intake was more balanced, magnesium and vitamin K intake remained low among the participants, which could jeopardise proper bone metabolism.

The multifactorial approach to osteoporosis must include not only nutrient supplementation but also the promotion of healthy lifestyle habits. The majority of participants (89.5%) did not consume alcohol, and 96.5% did not smoke (Table 2), which is highlighted as a positive point.

Practising physical activity requires attention; only 52.3% of the participants reported doing at least one hour of physical activity a week (Table 1). Most of those who exercised (78.8%) only did low-impact walking. The rest practised cycling (5.8%), pilates (7.7%), gym (3.8%) and yoga (3.8%). Walking is beneficial for cardiovascular health, but it may not be enough to prevent osteoporosis. Studies indicate that higher-impact physical activities, such as resistance exercises or weight lifting, are more effective at increasing or maintaining bone density, especially in people at higher risk of osteoporosis, as indicated by the Portuguese Society of Rheumatology and the American College of Sports Medicine [46,47].

The preventive approach should include healthy eating habits, regular exercise, adequate supplementation and specific strategies to reduce the risk of fractures by maximising peak bone mass and minimising bone loss from childhood onwards. Proper childhood nutrition is essential for optimising BMD, with special emphasis on the recommended daily intake of calcium, vitamin D, magnesium and vitamin K [17,43].

The data from this study regarding bone mineral density (BMD) in the spine revealed that 51.2% of the participants had osteopenia and 31.4% had osteoporosis, reflecting a significant prevalence of bone fragility among the study population. Furthermore, among the 44 participants with osteopenia, 11 had a medical diagnosis of osteoporosis and of the 27 diagnosed with osteoporosis, 9 had the condition confirmed by a doctor (Table 4). This study shows a statistically significant association between BMD assessment in the spine and confirmed medical diagnosis (*p* = 0.046), suggesting that the method effectively identifies cases of osteopenia and osteoporosis, complementing clinical assessments.

When assessing the BMD of the femur, the results showed a significant prevalence of osteopenia (43.0%) and osteoporosis (34.9%). Of the 37 participants with osteopenia in the femur, 8 had medical confirmation of osteoporosis and of the 30 with results of osteoporosis in the femur, 12 had the problem confirmed (Table 5). Thus, this study proves a statistically significant association between BMD in the femur by the REMS method and confirmed medical diagnosis (*p* < 0.01).

These data are in line with the aim of the study. They are in line with previously published studies, such as the female population studies by Amorim et al. (2021) and Di Paola et al. (2019), which concluded that the REMS method had a good ability to discriminate between normal and osteoporotic individuals, both in the spine and in the proximal femur in women [30,31]. The multicentre study by Giovanni et al. (2024) in the male population also demonstrated that it is an effective method, with a sensitivity and specificity of over 90% [48].

Furthermore, the cross-sectional study by Nowakowska-Płaza et al. (2021) revealed that radiofrequency echographic multispectrometry is a new densitometric method with significant diagnostic agreement with DXA for both sexes. This accuracy is acceptable regardless of the age and BMI of the users, which makes it especially useful for diagnosing osteoporosis in the elderly [29].

Our analysis also shows that the participants with normal spine and femur assessment results did not have doctors confirm their diagnoses, indicating the method’s good prediction, as seen in Table 4 and Table 5.

The lack of a significant association between BMD in the spine and femur and the occurrence of fractures (*p* = 0.278) can be attributed to the complexity of risk factors for fractures, including BMD but also falls, muscle weakness and lifestyle. The literature suggests that the risk of fractures is multifactorial; although the diagnosis of osteoporosis is based on BMD, the risk of fracture does not depend exclusively on it [49]. Studies have shown that factors such as age, hormonal status, physical activity and comorbid conditions play crucial roles in fracture susceptibility. Zhuang et al. (2020) showed that the risk of femur fracture increases significantly with age, even with constant BMD [50].

The results of the assessment of BMD in the spine showed no significant association with physical activity (*p* = 0.488), consumption of essential nutrients for bone health, such as vitamin D (*p* = 0.850), vitamin K (*p* = 0.801), calcium (*p* = 0.958), phosphorus (0.146), magnesium (*p* = 0.371) and the presence of pathologies (*p* = 0.837). The same was true for the assessment of BMD in the femur, which showed no significant association with physical activity (*p* = 0.854), the consumption of foods rich in vitamin D (*p* = 929), vitamin K (*p* = 0.297), calcium (*p* = 0.818), phosphorus (*p* = 0.419), magnesium (*p* = 0.949) and the presence of pathologies (*p* = 0.216).

REMS technology eliminates user positioning errors and artefacts, allowing for a more precise analysis of the trabecular bone. If the data collected are inaccurate, then the result cannot be obtained, requiring the examination to be repeated. ESCEO has recognised this advantage [26,29]. It is known that measuring BMD by DXA in the lumbar spine can result in an overestimation of BMD due to the degenerative changes of osteoarthritis, especially in adults and older people [24].

REMS technology eliminates user positioning errors and artefacts. The software can identify the portions of the signal referring to cartilage, cortical and trabecular bone. It selects only the trabecular portion, ensuring that the corresponding artefacts from the cortical bone do not interfere with the results. Once all the data have been analysed, the system checks that the number of interfaces detected corresponds to the spectral model of the trabecular bone and that they are sufficient to obtain a reliable diagnosis [28,31]. Suppose the area is considered non-diagnostic or the measurement does not provide signals of sufficient quality. In that case, the operator will not receive the examination result and must perform it again. This advantage has been recognised by ESCEO [26,29]. It is known that measuring BMD by DXA in the lumbar spine can result in overestimation of BMD due to degenerative changes from osteoarthritis, especially in adults and the elderly study stands out for its introduction of REMS technology in a primary care setting, exploring its potential as an innovative and accessible alternative for the early detection of osteoporosis [24].

REMS technology proves to be advantageous in this context. The study by Caffarelli et al. (2024) involved 500 users diagnosed with osteoarthritis who underwent BMD measurements at the lumbar spine and femoral sites using DXA and REMS. The results showed that the T-Score values in the lumbar spine by DXA were significantly higher than those obtained by REMS, especially in users with a higher degree of severity of osteoarthritis [32]. Studies by Caffarelli et al. in 2022 corroborate the above, which indicated that REMS technology could improve the diagnosis of osteoporosis in individuals with an apparent increase in BMD on DXA due to vertebral fractures or osteoarthritis in the lumbar spine [51].

Radiofrequency echographic multispectrometry is a new densitometric method that shows significant diagnostic agreement with traditional DXA. This method is particularly useful in diagnosing osteoporosis in the elderly, avoiding overestimation of BMD due to degenerative lesions. It can also be applied to specific groups, such as children and pregnant women, as it does not involve ionising radiation [51,52].

Early diagnosis is the key to adequate treatment of osteoporosis. To date, DXA is the gold-standard method, but some limitations are important, such as radiation dose, portability and high costs. These barriers do not allow it to be the true gold-standard technique or a screening tool at the primary care level [33,53].

This study suggests the effective integration of non-invasive technological tools in the early diagnosis of bone diseases in primary healthcare. It challenges the notion that bone mineral density assessment should rely exclusively on imaging centres equipped with large devices such as DXA, thus redefining diagnostic practice in the context of family health teams [30,51]. Ultrasound-based techniques do not involve exposure to radiation and represent a cheap, portable and increasingly reliable solution for assessing bone quality [53]

The literature review by Al Refaie et al. (2023) points out several advantages of REMS technology, including the possibility of assessing bone quality, overcoming limitations of DXA, allowing for periodic assessments without radiation in specific situations and offering a cost-effective and portable approach [54]. This reinforces the reliability of the proposed method, suggesting that it could be an effective tool for identifying cases of osteoporosis in primary care settings, especially where access to more sophisticated methods is limited [31].

REMS technology offers a more accessible and portable alternative without compromising diagnostic accuracy. This change suggests a new approach to the diagnosis and treatment of osteoporosis, centred on ease of use and the possibility of earlier interventions, which can reduce the costs and impacts of long-term therapies. Recent literature confirms that early identification of the population at risk can improve the quality of life of individuals and their social networks, as well as limiting the indirect costs associated with osteoporotic fractures in the working population [32,55].

Greater involvement among primary care teams in patient education significantly increases diagnosis and treatment rates [56,57]. The recent literature states that the creation of dedicated bone health teams in primary care significantly improves screening, diagnosis and treatment rates [57,58,59].

The growing use of portable devices in various areas, such as emergency units, intensive care units, operating theatres, long-term care units, and in the community, such as REMS technology, allows for a local and decentralised approach, increasing the efficiency and scope of osteoporosis diagnosis [60,61].

Community-based public health programmes that promote lifestyle changes effectively raise awareness and encourage preventative behaviours among at-risk populations [62]. However, despite established guidelines and recommendations, disparities in the diagnosis and treatment of osteoporosis persist, mainly due to economic and educational factors and access to health services [63].

The present study may be promising because it includes participants aged 30 and over and men. The survey by Caffarelli et al. (2022) highlighted the usefulness of REMS technology in assessing bone status in young women with anorexia, suggesting that this technique may be safe and effective for monitoring bone health over time [64]. Similarly, the study by Giovanni et al. (2024) evaluated the diagnostic accuracy of REMS technology in a male population, confirming its reliability in diagnosing osteoporosis in young men as well [48].

## 5. Limitations

Despite the promising results, this study has some limitations that should be considered. The sample consisted exclusively of Caucasian individuals, which may limit the generalisability of the results to more diverse populations. In addition, the small sample size may jeopardise the robustness of the conclusions.

The use of self-administered questionnaires to collect information on fracture history and lifestyle habits, such as physical activity, smoking and alcohol consumption, may introduce recall or response bias, affecting the accuracy of the data.

Another important limitation is the lack of specific software for analysing the frailty score and body composition. Assessment of the frailty index is a crucial tool for estimating fracture risk and providing valuable information for the management of osteoporosis. As a recently developed diagnostic indicator, it can help predict the imminent risk of fracture in the short term, benefiting individuals at risk of refracture and improving the early identification of high-risk categories, allowing for more effective preventive measures [26]. In addition, nutritional assessments based on the REMS method promote a more holistic approach to bone health. Collaboration between radiographers and nutritionists can be particularly beneficial, allowing for an exchange of information that potentially increases the effectiveness of public health interventions, especially in patients with obesity or anorexia.

In addition, this study has methodological limitations, the data are not compared with the results of dual-energy X-ray absorptiometry and there is no clinical data on osteoporosis and osteopenia, these data were collected in the questionnaire completed by the participants.

## 6. Conclusions

In conclusion, REMS technology effectively identifies cases of osteoporosis and osteopenia, representing a significant advance in detecting and monitoring osteoporosis. This technology is portable, does not use ionising radiation and is particularly suitable for vulnerable populations such as young people, pregnant women and the elderly. The portability of this equipment facilitates closer monitoring of users and improves access to diagnosis and adherence to treatment. This decentralised approach allows radiology technicians to move between different health units, with their integration into multidisciplinary teams, facilitating detection and diagnosis. Future studies are essential to explore the use of REMS in other demographic groups and better understand its role in diagnosing osteoporosis in primary healthcare.

## Data Availability

The original contributions presented in this study are included in the article. Further questions can be directed to the corresponding author(s).

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
