# Peer review of "Osteoporosis Evaluation by Radiofrequency Echographic Multispectrometry (REMS) in Primary Healthcare"

_diagnostics, 2025, doi:10.3390/diagnostics15070808_

Round 1
Reviewer 1 Report
Comments and Suggestions for Authors
The abstract and the introduction are well-written and conveys the main points effectively, but they would benefit from minor language refinements and additional context for clarity and completeness:
from ABSTRACT:
- The phrase "using radiofrequency echographic multi-spectrometry technology" is repeated twice. It could be simplified to "using REMS technology" after the first mention, as long as the acronym is clearly defined initially;
- The sentence starting with "Bone mineral density was assessed in 86 participants, using 172 scans of the lumbar spine and femur" could be more fluid if rephrased. For example, "Bone mineral density was assessed in 86 participants through 172 scans of the lumbar spine and femur..."
from INTRODUCTION:
- In the sentence "the ageing population caused by increasing life expectancy and decreasing mortality," "decreasing mortality" could be rephrased to "declining mortality rates" for better clarity.
- In "the cost of fractures in Portugal represents 5.6% of health expenditure, one of the highest figures in Europe," consider rewording it slightly: "the cost of fractures in Portugal accounts for 5.6% of health expenditure, one of the highest percentages in Europe."
- You mention that DXA is the gold standard in Portugal but has limitations (e.g., ionizing radiation). The contrast with REMS is well-explained, but it would be beneficial to mention specific advantages of REMS over DXA in more detail. For example, REMS does not just avoid radiation but also potentially offers faster or more convenient diagnostics, which would strengthen the argument for its use in primary care.
- The concluding sentence "This study aimed to contribute to the early detection of osteoporosis in primary healthcare using portable equipment and without ionising radiation" effectively sums up the study’s purpose, but you might expand a bit on the significance of "portable" and "without ionizing radiation" in primary healthcare settings. It could emphasize how these features can make screening more widely accessible and practical.
from Material and Methods:
the Figure 1, must be "figure 3": there is a mistake in the numeration.
form Discussion:
The discussion is well-written and covers all the necessary points, making it informative and persuasive. It connects the study's findings to the broader literature and provides valuable insight into the potential of REMS technology for early osteoporosis detection. However, some minor improvements can be made in the clarity of certain sections, expanding on the limitations, and emphasizing the practical application of the study’s findings. With these revisions, the discussion will be even stronger.
from CONCLUSION:
The conclusion is clear and concise, effectively summarizing the key findings of the study regarding REMS technology.
The references provided are sufficient and effectively support the argument presented throughout the study. They are well-connected to the various aspects of the research, including the effectiveness of REMS technology in diagnosing osteoporosis, its advantages over traditional methods, and the importance of early detection.
Author Response
Por favor, veja o anexo.
Obrigado
Ana Vieira

Reviewer 2 Report
Comments and Suggestions for Authors
General comments:
The aim of the study was the assessment of bone mineral density in two primary healthcare units with REMS, compared to the standard, defined as clinical diagnosis. The originality of the study is not stated anywhere.
There are two major drawbacks of the study:
- Lack of the DXA as the gold standard
- Lack of a clinical definition of osteopenia and osteoporosis.
Specific comments:
Introduction:
Q1. l. 31. When citing the WHO, the appropriate reference is missing. Two references in Portugese language are cited instead.
Q2. More details of the REMS technology should be provided. While emphasizing portability and non-ionization, the basic principles behind the technology and the interpretation of findings remain unclear. For instance, it is not explained what induces the different colors in the final map.
Methods:
Q3. How was the clinical diagnosis of osteopenia/osteoporosis defined?
Q4. l. 129. Check »visualise and visualise« - it appears to be a duplication.
Q5. l. 154 – 168: This section should be moved to the Figure 3 Legend. Also, ensure that Figure captions are annotated appripriately.
Q6. Figure1Final report: rename this to Figure 3.
Q7. l. 226, Table 4.
In the last column of the »osteoporosis confirmed by a doctor«, the sum 66+20 should be 86 and not 27, as stated.
In the last column of »had a bone fracture« the sum should be 86. Why is it only 80?
Same question applies also for the Table 5, please verify.
Discussion:
Q8. The statement of the portability and non-ionizing concept of REMS is unnecessarily repeated multiple times.
Q9. l. 346. How does REMS eliminate user positioning errors and artefacts? Please clarify.
Q10. l. 361-365. This paragraph is more appripriate in the Introduction or the beginning of the Discussion.
References:
Q11. In general, 65 references are excessive for a paper of this type.
Q12. The first seven references are in Portugese language. In general, in peer-reviewed journals, references should be in English.
Author Response
Please see the attachment.
Thank You
Ana Vieira

Round 2
Reviewer 2 Report
Comments and Suggestions for Authors
Most revisions are acceptable, thank you, however the main two major shortcomings in methodology remain:
- The absence of DXA as the gold standard
- The lack of a clinical definition of osteopenia and osteoporosis
To my knowledge, osteoporosis cannot be reliably diagnosed using a questionnaire, let alone to the extent that it could be used as a gold standard for evaluating other diagnostic methods such as REMS. If a questionnaires were diagnostic, what would be the need for DXA?
Author Response

(The authors gave the same response as above.)

Round 3
Reviewer 2 Report
Comments and Suggestions for Authors
In the latest letter to the Editor, the authors state several reasons for not using DXA as a gold standard, which are correctly explained.
However, I would still expect the authors to explain the origin of the questionnaire and, above all, on what basis this questionnaire has been considered a gold standard for the evaluation of REMS.
Author Response

(The authors gave the same response as above.)
